# Mechanisms of Cefiderocol Resistance in Carbapenemase-Producing Enterobacterales: Insights from Comparative Genomics

**DOI:** 10.3390/antibiotics14070703

**Published:** 2025-07-12

**Authors:** Alexander Tristancho-Baró, Ana Isabel López-Calleja, Ana Milagro, Mónica Ariza, Víctor Viñeta, Blanca Fortuño, Concepción López, Miriam Latorre-Millán, Laura Clusa, David Badenas-Alzugaray, Rosa Martínez, Carmen Torres, Antonio Rezusta

**Affiliations:** 1Clinical Microbiology Laboratory, Miguel Servet University Hospital, 50009 Zaragoza, Spain; ailopezcal@salud.aragon.es (A.I.L.-C.); amilagro@salud.aragon.es (A.M.); mparizas@salud.aragon.es (M.A.); vvinneta@salud.aragon.es (V.V.); bmfortuno@salud.aragon.es (B.F.); clopezgo@salud.aragon.es (C.L.); arezusta@salud.aragon.es (A.R.); 2Research Group on Difficult to Diagnose and Treat Infections, Institute for Health Research Aragon, Miguel Servet University Hospital, 50009 Zaragoza, Spain; mlatorre@iisaragon.es (M.L.-M.); lclusa@iisaragon.es (L.C.); dbadenas@iisaragon.es (D.B.-A.); 3Area of Biochemistry and Molecular Biology, One Health-UR Research Group, University of La Rioja, 26006 Logroño, Spain; carmen.torres@unirioja.es; 4Infectious Diseases Department, Miguel Servet University Hospital, 50009 Zaragoza, Spain

**Keywords:** carbapenem-resistant enterobacterales, comparative genomics, antimicrobial resistance, whole-genome sequencing

## Abstract

**Background/Objectives**: Cefiderocol is a novel siderophore cephalosporin with potent in vitro activity against a broad spectrum of Gram-negative bacteria, including carbapenemase-producing Enterobacterales (CPE). However, the recent emergence of resistance in clinical settings raises important concerns regarding its long-term effectiveness. This study aims to investigate the genomic determinants associated with cefiderocol resistance in CPE isolates of human origin. **Methods**: Comparative genomic analyses were conducted between cefiderocol-susceptible and -resistant CPE isolates recovered from human clinical and epidemiological samples at a tertiary care hospital. Whole-genome sequencing, variant annotation, structural modelling, and pangenome analysis were performed to characterize resistance mechanisms. **Results**: A total of 59 isolates (29 resistant and 30 susceptible) were analyzed, predominantly comprising *Klebsiella pneumoniae*, *Escherichia coli*, and *Enterobacter cloacae*. The most frequent carbapenemase gene among the resistant isolates was *bla*_NDM_, which was also present in a subset of susceptible strains. The resistant isolates exhibited a significantly higher burden of non-synonymous mutations in their siderophore receptor genes, notably within *fecR*, *fecA*, *fiu*, and *cirA*. Structural modelling predicted deleterious effects for mutations such as *fecR:*G104S and *fecA:*A190T. Additionally, porin loss and loop 3 insertions (e.g., GD/TD) in OmpK36, as well as OmpK35 truncations, were more frequent in the resistant isolates, particularly in high-risk clones such as ST395 and ST512. Genes associated with toxin–antitoxin systems (*chpB2*, *pemI*) and a hypothetical metalloprotease (group_2577) were uniquely found in the resistant group. **Conclusions**: Cefiderocol resistance in CPE appears to be multifactorial. NDM-type metallo-β-lactamases and missense mutations in siderophore uptake systems—especially in those encoded by *fec*, *fhu*, and *cir* operons—play a central role. These may be further potentiated by alterations in membrane permeability, such as porin disruption and efflux deregulation. The integration of genomic and structural approaches provides valuable insights into emerging resistance mechanisms and may support the development of diagnostic tools and therapeutic strategies.

## 1. Introduction

Cefiderocol (CFD) is a novel cephalosporin that combines structural motifs similar to those found in ceftazidime and cefepime, namely dimethyl, oxime, and pyrrolidinium groups, which confer stability against a wide range of beta-lactamases, including carbapenemases. It features a chlorocatechol moiety in its C-3 side chain, enabling it to chelate trivalent iron ions, mimicking siderophores and leveraging bacterial iron uptake mechanisms to actively penetrate the periplasmic space, where it exerts its antibacterial activity predominantly through the inhibition of penicillin-binding protein (PBP) 3 [1]. Consequently, it has emerged as a promising therapeutic alternative for infections caused by difficult-to-treat Gram-negative bacteria [2]. However, since its clinical approval in 2019, resistant isolates and the emergence of resistance during treatment have been reported [3,4], underscoring the need to characterize its underlying mechanisms to optimize its use and prevent the spread of resistant strains, thus prolonging the antibiotic’s clinical utility.

The main resistance mechanisms identified to date involve alterations in siderophore receptors, specific beta-lactamase variants, mutations in PBP3, and changes in membrane permeability [5]. Evidence supporting these findings are derived from isogenic mutant construction assays, but genomic analyses of clinical isolates are still limited. Although observational studies suggest that resistance to CFD remains low, the growing number of resistance reports is concerning [6,7]. This study aims to compare the resistome of two collections of CPE of human origin, respectively, CFD-resistant and CFD-susceptible, in order to explore the possible genomic determinants of CFD resistance.

## 2. Results

### 2.1. Isolate Selection and Identification

From 1 January 2022 to 31 December 2024, a total of 172 non-duplicate (first isolate per patient) CPE isolates were identified from clinical and epidemiological samples at the microbiology laboratory of the Miguel Servet University Hospital (HUMS) in Zaragoza, Spain. The most prevalent species were *Klebsiella pneumoniae* complex, *Escherichia coli*, *Citrobacter* spp., and *Enterobacter cloacae* complex, accounting for 51%, 19%, 13%, and 10% of cases, respectively. Of these, 63 strains (~36%) were tested for CFD susceptibility by one of the accepted methods listed in the inclusion criteria. This resulted in 31 susceptible (CFD-S), 30 resistants (CFD-R), and 2 within the area of technical uncertainty that were excluded from further analysis.

After phenotypic evaluation, identification was confirmed prior to sequencing to ensure culture purity using matrix-assisted laser desorption ionization–time of flight mass spectrometry (MALDI-TOF MS). Following whole-genome sequencing (WGS), two strains—one from each group—were excluded due to an N-50 sub-threshold value and contamination, respectively. The contaminated sample contained sequences belonging to *Citrobacter portucalensis* (80%) and *Citrobacter cronae* (20%), both belonging to the *Citrobacter freundii* complex. Given that the sequencing depth was near the lowest acceptable limit, no filtering was applied, and the isolate was entirely excluded from this study. All remaining strains met quality control thresholds.

Consequently, the final cohort included 29 CFD-R and 30 CFD-S isolates. The species distribution based on WGS was as follows: resistant group: *K. pneumoniae* (n = 15), *E. coli* (n = 9), *Enterobacter hormaechei* (n = 2), *Enterobacter asburiae* (n = 1), *Enterobacter kobei* (n = 1), *C. portucalensis* (n = 1), and *Providencia stuartii* (n = 1); susceptible group: *K. pneumoniae* (n = 16), *Klebsiella variicola* (n = 1), *E. coli* (n = 5), *E. hormaechei* (n = 2), *C. portucalensis* (n = 1), *Citrobacter koseri* (n = 1), *P. stuartii* (n = 1), *Providencia hangzhouensis* (n = 1), *Morganella morganii* (n = 1), and *Serratia sarumanii* (n = 1).

### 2.2. Origin of Isolates and Antimicrobial Susceptibility Testing

Approximately 80% of the samples were obtained from male patients. The median age at the time of sample collection was 46 years, ranging from 7 to 88 years. No significant age differences were observed between the resistant and susceptible groups.

A total of 59% of the isolates were recovered from epidemiological samples, primarily from rectal or triple-site swabs collected as part of the hospital’s “Zero Resistance” programme [8]. The distribution of epidemiological samples was comparable between the CFD-R and CFD-S groups (*p* > 0.05). The remaining samples were of clinical origin, with urine and wound exudate being the most prevalent sample types, accounting for 16.9% and 13.5%, respectively, with no significant differences in sample type distribution between the groups. The fact that an isolate was obtained from an epidemiological sample does not preclude the possibility that the patient developed a clinical infection later during hospitalization or after discharge. Three isolates originated from invasive samples (ascitic fluid, prosthetic joint material, and blood).

The overall rate of antibiotic resistance was high in both groups. All isolates showed non-susceptibility to ceftazidime and cefepime. Resistance to carbapenems (ertapenem, imipenem, and meropenem) was 89.8%, 89.8%, and 81.4%, respectively. Approximately three-quarters of the isolates were resistant to the second- and third-generation fluoroquinolones ciprofloxacin and levofloxacin. Aminoglycoside resistance ranged from 54.2% for amikacin to 91.5% for tobramycin. The resistance rates for the last-resort antibiotics fosfomycin, tigecycline, and colistin were 37.3%, 35.6%, and 18.6%, respectively. Furthermore, the resistance rates were 100% and 78% for the new-generation antibiotics ceftolozane–tazobactam and ceftazidime–avibactam. These resistance patterns were interpreted in light of the intrinsic resistance profiles of the Enterobacterales species evaluated, including chromosomal AmpC expression in certain species. The fosfomycin results were restricted to *E. coli*, and tigecycline results were interpreted based on *E. coli*-specific breakpoints. The full antimicrobial profile is shown in Appendix A.

### 2.3. Sequence Types, Resistome, and Plasmid Characterization

A high diversity of sequence types (STs) was observed among the global set of isolates. Among the CFD-R *K. pneumoniae* isolates, ST395 was the most prevalent, while ST23 and ST512 were exclusive to this group. Conversely, ST147 was the most common ST in the CFD-S group, though these differences were not statistically significant. No predominant STs or statistically significant differences were observed in the other bacterial species.

Both groups exhibited complex resistomes, including Ambler class A, B, and D carbapenemases [9]. Among the CFD-R group, 38 carbapenemase genes were detected, including isolates with co-existing metallo- and serine-carbapenemases. Genes encoding NDM-type enzymes were the most frequent, specifically *bla*_NDM-1_ (n = 13), followed by *bla*_KPC-3_ (n = 6) and *bla*_NDM-5_ (n = 5). Notably, *bla*_NDM-1_
*+ bla*_OXA-48-like_ co-occurrence was identified in seven isolates. In the CFD-S group, genes encoding VIM-type carbapenemases were significantly more common (*p* < 0.05), particularly *bla*_VIM-1_ (n = 12), followed by *bla*_NDM-1_ (n = 10). Only one isolate carried *bla*_KPC-3_, and the only combination observed was *bla*_NDM-1_
*+ bla*_OXA-48_, found in five isolates.

Genes encoding extended-spectrum beta-lactamases (ESBLs) were detected in half of the collection, primarily *bla*_CTX-M-15_, with no significant differences observed between the groups. However, *bla*_CTX-M-55_ and *bla*_CTX-M-9_ were exclusively found in the CFD-R and CFD-S groups, respectively. Plasmid-mediated AmpC-type beta-lactamases (*bla*_CMY_ alleles) were exclusively detected in the CFD-R group (*p* < 0.05).

Mutations in *gyrA* and *parC* were more common in the CFD-R group, particularly those related to D87N and S83I changes in GyrA for *E. coli* and *K. pneumoniae*, respectively. Numerous genes encoding acetyltransferases, nucleotidyltransferases, and phosphotransferases were found in both groups. Notably, the *mcr-10.1* and *mcr-1.1* alleles were detected in one *E. asburiae* isolate (resistant group) and one *E. coli* isolate (susceptible group), respectively.

At least one carbapenemase-encoding plasmid was successfully reconstructed in 86% of the CFD-R isolates. The average plasmid size was 140 Kb (range: 4.3–353 Kb), belonging to various incompatibility groups, mainly *IncF* and *IncC*. Conjugative plasmids accounted for 72%, with MOBH and MOBF being the most frequent relaxases. Five isolates harboured at least two plasmids encoding carbapenemases. In the CFD-S group, 90% of the carbapenemase-harbouring plasmids were reconstructed, with at least two plasmids detected in five isolates. The average plasmid size was approximately 130 Kb, with *IncL* and *IncF* being the most common incompatibility groups. Conjugative plasmids represented 74% of the total, and MOBP was the most frequent relaxase.

Table 1 summarizes the main microbiological and epidemiological features of the included isolates, showing the distribution of the species, sample origin, carbapenemase type, and cefiderocol susceptibility profile.

### 2.4. Mutation Analysis

#### 2.4.1. Genes Involved in Iron Metabolism

Functional annotation enabled the selection of genes associated with iron uptake and regulation at the cellular level. Using this list, genes bearing missense mutations were filtered in both groups based on nucleotide variant annotation. In the CFD-R group, 855 unique missense mutations were identified across 59 genes, with the *fhu*, *fep*, and *fec* operons showing the highest mutational burden.

Subsequently, mutations exclusive to the CFD-R group were selected to focus on biologically plausible mechanisms of CFD resistance. A total of 119 unique mutations were identified in seven key genes: *fecB*, *fes*, *fiu*, *cirA*, *fhuC*, *nfeF*, and *fhuF*. Overall, the highest numbers of mutations were observed in *fhuF*, *fes*, and *fiu* with 82, 69, and 35 variants, respectively. Noteworthy high-prevalence missense mutations among the isolates included V65A in *fhuF*, I57S in *fes*, and G465D in *cirA*. To determine whether the mutational burden in these genes represents a distinguishing feature of the CFD-R group, the total number of missense mutations per gene was compared between groups, revealing statistically significant differences in *fecB*, *fes*, *fiu*, *cirA*, *fhu*, *nfeF*, and *fhuF* (*p* < 0.05).

After identifying the loci and differential mutations, we assessed whether these changes might indicate a loss of protein function by comparing the folding energy difference between the wild-type and mutant proteins. Due to the need for reference 3D models, this analysis was limited to *E. coli*, whose proteome is well characterized. Mutations were classified as deleterious (>1.6 kcal/mol), intermediate (0.5–1.5 kcal/mol), or neutral (<0.5 kcal/mol) [10]. In total, 182 nucleotide variants (associated to amino acid change) were found across 40 genes, of which 40 were considered intermediate and 28 deleterious (Figure 1). Among the deleterious mutations (18 loci), *fecR*:G104S, *fecA*:A190T, and *fiu*:R212H were those with the highest delta energy gap (ΔΔG) values of 11.7, 5.7, and 5.3 kcal/mol, respectively (Figure 2). The average ΔΔG values per gene were also calculated, with *fecR*, *fecA*, and *sdhB* exhibiting the highest averages of 4.1, 1.8, and 1.4 kcal/mol, respectively.

#### 2.4.2. Penicillin-Binding Proteins (PBPs)

Using a similar approach, exclusive nucleotide mutations were filtered for PBP1a, 1b, and 2–7, which correspond to the *mrcA*, *mrcB*, *mrdA*, *ftsI*, *dacB*, *dacA*, *dacC*, and *pbpG* loci in *E. coli*. A total of 94 mutations were found across seven of these eight loci in the CFD-R group. Of these, 21 variants (distributed across PBP1a–PBP4) were exclusive to this group. PBP4 and PBP3 showed the highest number of unique variants, accounting for seven and five, respectively.

Regarding structural impact, all PBP mutations were predicted to be neutral, except for *mrcA*:G414D, which showed a ΔΔG of 2.9 kcal/mol.

#### 2.4.3. Efflux Pumps

Efflux pump systems are frequently associated with multidrug resistance. We catalogued all the exclusive mutations in two major efflux systems in Enterobacterales, AcrAB-TolC and OqxAB, including their associated regulatory and structural components.

A total of 128 unique nucleotide variants were identified, of which 97% belonged to the AcrAB-TolC system. Approximately 65% of these mutations were located in accessory genes such as *acr*D-F and *acr*R. Notably, 44% of the mutations were found in the *acr*R regulatory gene.

#### 2.4.4. Allelic Variants in Carbapenemases and Class C Beta-Lactamases

Certain aminoacidic changes in beta-lactamase genes can alter their hydrolytic profiles, conferring enhanced resistance. These changes have been primarily documented in carbapenemases and class C beta-lactamases, sometimes resulting in distinct allelic variants [11].

In the resistant group, the sequences of *bla*_KPC-2_, *bla*_KPC-3,_
*bla*_NDM-1,_
*bla*_NDM-5_, *bla*_VIM-1_, *bla*_OXA-48_, *bla*_OXA-181_, and *bla*_OXA-244_ were aligned against their respective references. All the aligned sequences showed 100% identity with the reference.

The analysis of class C beta-lactamases included both chromosomal and plasmid-mediated alleles: *bla*_EC-8_, *bla*_EC-14_, *bla*_EC-15_, *bla*_ACT-40_, *bla*_ACT-52_, *bla*_ACT-55_, *bla*_MIR-3_, *bla*_CMY-4_, *bla*_CMY-6_, *bla*_CMY-13_, *bla*_CMY-16_, *bla*_CMY-42_, and *bla*_CMY-181_. After filtering out variants shared with the CFD-S group, mutations were detected in the *bla*_EC_ alleles of seven *E. coli* isolates, as well as a duplication in *bla*_ACT-55_ from *E. hormaechei*. Notably, *bla*_EC-8_ was exclusive to the CFD-R group.

The enhanced hydrolytic activity in AmpC beta-lactamases has often been linked to mutations within the Ω-loop [12,13]. We identified the R248C aminoacidic change in EC-14 and EC-15, as well as N201T, P209S, and S298I changes in all EC-8 isolates. Of these, P209S is located close to α-helix 8, which configures the enzyme’s active site [14]. In silico structural modelling predicted a root mean square deviation (RMSD) of 0.008 Å for the mutated protein compared to the wild type, indicating negligible structural disruption. Other aminoacidic changes outside the Ω-loop included Q23K, P110S, and A367T in EC-14/15, and T4M, S102I, Q196H, and T367A in EC-8.

#### 2.4.5. Porin Loss

Porin loss may act as a complementary mechanism in antibiotic resistance, especially OmpK35 and OmpK36 in *K. pneumoniae*. Out of the CFD-R isolates, 12 were found to harbour mutations resulting in truncated OmpK35 proteins with a predicted length of less than 75% of the wild-type sequence. Regarding mutations in loop 3 of OmpK36, GD deletion was identified in eight isolates and TD deletion in two. Additionally, one isolate exhibited a premature truncation of OmpK36, resulting in a peptide comprising less than 20% of the expected full-length protein. In contrast, only eight isolates in the CFD-S group showed any form of alteration in OmpK35 or OmpK36, representing a significantly lower frequency compared to the CFD-R group (*p* < 0.05). Moreover, the average predicted length of OmpK35 was higher in the CFD-S group (72.7%) compared to the resistant group (50.9%) (*p* < 0.05) (Figure 3).

#### 2.4.6. Pangenome Analysis

To enable a more comprehensive and unbiased investigation into the potential genes associated with cefiderocol resistance, a presence/absence analysis based on the pangenome of the isolates was conducted, comparing the CFD-R and CFD-S groups. Due to the number of isolates available for each species, this analysis was restricted to *E. coli*, *K. pneumoniae*, and *E. hormaechei*. Only group-exclusive loci—defined as genes present in all isolates of one group and absent from all isolates of the other—were considered.

Among the three species, *K. pneumoniae* exhibited the largest pangenome, comprising 12,555 genes, followed by *E. coli* with 11,839 and *E. hormaechei* with 10,650. A similar distribution was observed for their core genomes. Notably, in both *K. pneumoniae* and *E. coli*, cloud genes—those present in fewer than 15% of the isolates—accounted for 47.3% and 49.9% of their pangenome, respectively. This highlights substantial intraspecies variability and supports the presence of an open pangenome within the collection, consistent with findings from previous studies [15]. Similarly, in *E. hormaechei*, 80.8% of the pangenome consisted of shell genes (present in 15–95%of isolates), although this may reflect the limited number of isolates available for this species.

Regarding the presence/absence analysis, no group-exclusive genes were identified in *K. pneumoniae* or *E. hormaechei*. However, in *E. coli*, three loci were exclusive to the CFD-R group and one to the CFD-S group. The resistant-exclusive genes included *chpB2*, *pemI*, and a hypothetical protein internally annotated as group_2577 (Figure 3), which exhibited 100% identity and coverage with a metalloprotease identified in the metallo-beta-lactamase-producing *E. coli* strain EC_BZ_10 from Italy [16]. The locus exclusive to the CFD-S group was also a hypothetical protein, designated *group_2368*, with high sequence homology to a lipoprotein from *E. coli* O139:H28 (strain E24377A/ETEC).

Table 2 summarizes exclusive variants found in the resistant group by locus and resistance mechanism.

## 3. Discussion

This study compared the genomic differences between a collection of cefiderocol-resistant isolates and a set of susceptible isolates with similar characteristics.

The main CPE species or CPE species complexes included in this study were *K. pneumoniae*, *E. coli*, and *E. cloacae*. Additional isolates included *Citrobacter* spp., *Providencia* spp., *Morganella morganii*, and *Serratia sarumanii*, the latter three being exclusive to the susceptible group. All isolates were obtained from human samples, collected for clinical or epidemiological purposes, which strengthens the clinical relevance of the findings.

The higher rate of aztreonam susceptibility may be attributed to the high prevalence of metallo-beta-lactamases among the isolates. However, the co-occurrence of other beta-lactamases from classes A and C limits aztreonam’s effectiveness and confines its susceptibility rate to approximately 25%, thereby restricting its utility as a monotherapy option. In this context, non-beta-lactam antibiotics such as amikacin and colistin may gain importance as components of combination therapies in the absence of alternative treatment options, although their use is limited by a high incidence of adverse effects [17].

The isolation of three strains from invasive clinical samples highlights the urgent need for effective therapeutic alternatives. Cefiderocol resistance, combined with the multidrug-resistant profiles exhibited by most of the isolates, significantly reduces the likelihood of effective treatment and worsens patient prognosis [18].

### 3.1. Mutations in Genes Related to Iron Transport Systems

By mimicking the biological function of siderophores, CFD is highly dependent on intracellular iron transport systems in order to exert its antimicrobial activity [1]. Therefore, a reduction in the expression or function of these proteins may contribute to its resistance.

The most notable differences between the two groups include a higher mutational burden in *fecB*, *fes*, *fiu*, *cirA*, *fhu*, *nfeF*, and *fhuF*. These genes have been previously reported as potential contributors to cefiderocol resistance [19,20,21,22,23], primarily in experimental models where they led to an increase in minimum inhibitory concentrations of 2- to 16-fold from baseline [24]. In particular, *cirA* has been identified in clinical isolates exhibiting resistance development during treatment [25]. Our findings suggest that several mutations induced under laboratory conditions indicated in the referred study might indeed have a real-world impact in clinical, human-derived isolates.

Many of these genes are functionally interconnected, so a single mutation can impair the entire system. Additionally, the number of mutations does not always predict functional loss. To address this issue, the energy difference (ΔΔG) associated with 182 deduced aminoacidic changes across 40 loci in resistant *E. coli* isolates was evaluated as an indirect indicator of loss of protein function. A total of 28 variants were predicted to be deleterious, affecting genes such as *efeO*, *fecA*, *fecC*, *fecR*, *fepB*, *fepD*, *fepG*, *fes*, *fhuB*, *fhuC*, *fhuD*, *fhuE*, *fhuF*, *fiu*, *glcF*, *nfeF*, *sdhB*, and *yfaE* (see Figure 4). These mutations could result in misfolded or unstable proteins subject to degradation [26]. Although these mutations may have arisen as a result of environmental selective pressures or stochastic events, their absence in the CFD-S group likely suggests prior exposure to cefiderocol or other antibiotics, or the selection of minor heteroresistant subpopulations following exposure to the molecule. Such mutations could confer adaptive advantages over other bacterial populations in the context of host adaptation, thereby facilitating the development of infection or colonization.

For instance, in the *fec* operon, key functions such as the activity of the outer membrane siderophore (*fecA*) and the positive regulation of operon expression (*fecR*) were compromised by mutations associated with energy differences of 5.7 and 11.7 kcal/mol, respectively. Another molecular key in the bacterial iron uptake systems, the *fhu* operon, which encodes a high-affinity, energy-dependent transport system, also appeared significantly affected by deleterious mutations impacting the following key components of the system: the outer membrane receptor (*fhuE*), the periplasmic and inner membrane transporters (*fhuD*, *fhuB*), and the ATPase component (*fhuC*). These alterations may compromise the functionality of the iron uptake machinery, potentially impairing cefiderocol internalization but also limiting bacterial fitness, especially in iron-depleted environments. This fitness impact may be compensated by degenerate pathways [27] and the overexpression of other iron uptake mechanisms with less affinity for CFD. Further analysis involving gene expression, transcriptomics, proteomics, and/or metabolomics are required to validate this hypothesis and further elucidate the role of siderophores’ function and regulation in the context of human infections and antibiotic exposure.

### 3.2. Presence, Mutation, and Overexpression of Specific Beta-Lactamases

The presence of certain enzymes has been associated with increased MICs and an increased likelihood for resistance development. In this regard, a higher prevalence (42–59%) of NDM-type metallo-beta-lactamases has been reported among cefiderocol-resistant isolates in several studies [28,29,30]. In line with these findings, *bla_NDM_* was more frequently detected in the resistant group, although it was also identified in approximately one-third of the CFD-S isolates. This observation likely reflects the local epidemiology of CPE in our setting [31] and the fact that susceptibility to cefiderocol is more often tested in isolates exhibiting extensive resistance profiles.

Moreover, resistant isolates more frequently harboured multiple carbapenemase genes. Various *blaKPC* allelic variants (e.g., *bla*_KPC-31_, *bla*_KPC-33_, *bla*_KPC-41_, *bla*_KPC-50_, *bla*_KPC-25_, *bla*_KPC-29_, *bla*_KPC-44_, *bla*_KPC-121_, *bla*_KPC-203_, *bla*_KPC-109_, and *bla*_KPC-216_) have been associated with cefiderocol resistance in prior studies [32,33]. However, none of these variants were detected in our cohort, which may be correlated with geographical or clonal expansion patterns. Likewise, we did not observe cross-resistance to ceftazidime–avibactam in isolates carrying class A carbapenemases, nor did we detect *bla*_OXA-427_, another carbapenemase with potential cefiderocol hydrolytic activity [34]. These allelic variants, although less frequent, should be taken into account due to their clinical impact, highlighting the benefits of whole-genome sequencing for the surveillance of multidrug-resistant microorganisms. Notable examples of CFD resistance mediated by class C beta-lactamases include A292_L293del in EC, A313P and A292_L293del in ACT, and A114E, Q120K, V211S, and N346Y in CMY-2, as well as the presence of *bla*_CMY-186_ in *K. pneumoniae* [19]. In our dataset, exclusive mutations in CFD-R were found in several allelic variants (i.e., EC-8, EC-14, EC-15, and ACT-55), none of which matched previously reported resistance-associated mutations (see Table 2). Of particular interest is the P209S substitution located within the Ω-loop, a region known to influence the hydrolytic profile of beta-lactamases and confer resistance to agents such as ceftazidime–avibactam [12]. However, its minimal impact on the three-dimensional configuration of the enzyme suggests that its functional effect is likely negligible. EC-8 was found exclusively in the resistant group, but its low prevalence limited any statistically meaningful association. Experimental studies are needed to assess the impact of these and other mutations on cefiderocol and other antibiotic susceptibility profiles.

It is noteworthy that *bla*_NDM_ and *bla*_KPC-3_ were more common among the resistant isolates in our study, reflecting the importance of understanding the local and regional epidemiology of CPE.

No additional beta-lactamases previously associated with cefiderocol resistance in Enterobacterales—such as CTX-M-27, PER, SPM-1, BEL, or extended-spectrum SHV variants—were detected [5,19].

The selection of mutations in β-lactamases that confer resistance to CFD may become increasingly relevant if such mutated variants are located within mobile genetic elements as they may confer cross-resistance to other last-resort beta-lactams such as ceftazidime–avibactam and ceftolozane–tazobactam. Their characterization through WGS is essential to inform the development of rapid, simplified molecular diagnostic tools for implementation in clinical microbiology laboratories.

### 3.3. Alterations in Penicillin-Binding Proteins (PBPs)

Sato and colleagues evaluated the impact of insYRIN and insYRIK insertions in the protein encoded by *fstI* (PBP3) (encoding PBP3) in *E. coli* isolates, reporting a two-fold increase in the cefiderocol MIC [35]. Similarly, Price et al. identified the insYRIN insertion during the genomic characterization of a collection of cefiderocol-resistant *E. coli* isolates, observing a comparable elevation in MIC values [3]. Structural alterations in PBP-3 may reduce its affinity for CFD and thereby confer resistance.

In our study, mutations exclusive to the CFD-R group were identified in *mrcA*, *mrcB*, *ftsI*, and *dacB* (see Table 2). None of these were predicted to significantly impair the three-dimensional structure of the protein. Unlike siderophore-related mechanisms, resistance in this context is not necessarily associated with the loss of function, but rather with the altered binding affinity of the antibiotic to the target protein.

In this regard, the *ftsI* variants I332V and E349K, found in our study, are involved in a potential role in the active site conformation of the protein. It is plausible that only mutations capable of altering antibiotic affinity without compromising protein function would be selectively retained at the population level. Additional experimental studies capable of assessing the CFD affinity to these *ftsI* allelic variants are warranted to confirm their impact on CFD susceptibility.

No isolates in our collection harboured insYRIN or insYRIK insertions.

The selection of one or more of these mutations may be related to the environment in which the isolate was found and its prior exposure to CFD or other antibiotics, as well as to additional factors, including the patient’s medical history and current clinical condition, the bacterial species, its sequence type, and the intraspecies biological variability of each isolate.

### 3.4. Alterations in Permeability and Active Efflux

Deletions in the outer membrane porins OmpK35 and OmpK36 have been associated with increased cefiderocol MICs in *K. pneumoniae* [36]. Additionally, aminoacidic changes in OmpK36 have been described in high-risk clones, leading to pore constriction and reduced permeability to multiple antibiotics [37]. The marginal yet detectable impact of this mechanism on cefiderocol resistance is supported by the higher prevalence of such mutations (e.g., insGD, insTD, or large deletions) in the *K. pneumoniae* isolates from the resistant group, in line with findings reported by Simner et al. [36]. This trend was especially pronounced in sequence types ST395 and ST512, which were prevalent in our collection and are known for their pronounced virulence and resistance profiles.

Limited evidence is available regarding the specific role of the multidrug transporters AcrAB-TolC and OqxAB in CFD resistance [38]. In our study, a high frequency of mutations in the *acrR* regulatory gene—part of the AcrAB efflux system—was observed in the resistant group, suggesting the potential hyperactivation of this system.

These mutations may confer resistance by increasing the efflux of cefiderocol from the periplasmic space, thereby preventing its inhibition of PBP3 and ultimately allowing continued peptidoglycan synthesis, similar to the mechanism observed with other antibiotics in the same class. Such efflux pumps are non-specific and exhibit an affinity for multiple substrates, including antibiotics from various families. The presence of these mutations in non-specific pathways could enhance resistance when occurring alongside others in siderophores or β-lactamases. All mutations were absent in the susceptible isolates.

### 3.5. Pangenome Analysis

Finally, whole-genome association approaches, such as those proposed by Mosquera-Rendón et al., constitute a valuable strategy for generating hypotheses [39]. In this regard, our analysis identified genes with predicted endoribonuclease activity that are exclusively present in the CFD-R group and are linked to type II toxin–antitoxin systems (*chpB2* and *pemI*), the latter also being associated with plasmid replication stability. Additionally, we identified two hypothetical coding sequences (CDSs), one exclusive to the resistant group and the other to the susceptible group, with putative functions consistent with a metalloprotease and a lipoprotein, respectively, based on homology analyses.

These loci may serve as surrogate markers of the resistant phenotype in some species or clonal lineages, proven that these findings are validated in larger cohorts, opening new possibilities for the development and implementation of targeted molecular methods that are more cost-effective and focused on the early detection of high-risk clones with a substantial likelihood of CFD resistance. This approach would be particularly valuable in critical hospital areas.

## 4. Limitations and Future Directions

One of the main limitations of this study is its lack of parental (isogenic) strains, which precludes direct comparative genomics and limits the establishment of causal relationships, as resistance-associated mutations were identified and filtered out based on isolates with distinct genomic backgrounds. However, the groups were balanced both in overall count and species distribution. Moreover, the inclusion of multiple species adds biological diversity to the investigation of resistance mechanisms. Although the absence of putative resistance determinants in the susceptible group does not imply causality, it serves as a useful filter to exclude findings that are not likely to be related to resistance.

Similarly, other important limitations of this study include its relatively small sample size and the lack of clinical data on CFD exposure in the patients from whom the isolates were obtained, which hindered the assessment of the selective pressure that led to resistance-associated mutations. However, due to the intrinsic characteristics of the project, it was not feasible to increase it. Nonetheless, this work lays the foundation for a well-characterized strain collection that could be expanded in the future through new projects or collaborations with other institutions.

Searching for beta-lactamase mutations using draft genomes may fail to detect the low-frequency allelic variants not represented in the final assembly. However, mapping-based approaches pose additional challenges in this context, as these genes are often located on plasmids and complete reference sequences are lacking for many species. Moreover, low-frequency variants in genes that individually exert minor effects on antibiotic susceptibility are unlikely to alter group-level outcomes.

Inter-species genomic variability is a critical factor to consider when interpreting the findings of this study. Tracing recombination events across species may provide insight into the origin and potential environmental dissemination of these resistance mechanisms. In our study, we attempted to mitigate the impact of inter-species variability by performing species-specific analyses whenever the sample size and the mechanism under investigation allowed. Nevertheless, the inherent complexity of investigating resistance mechanisms in multi-species collections remains evident, particularly when the mechanisms involved are multifactorial in nature.

## 5. Materials and Methods

The primary data regarding the bacterial species, carbapenemase type, antimicrobial susceptibility profiles, and sample types from which the isolates were obtained were extracted from the Laboratory Information System (LIS) (SIGLO v2, Horus Software S.A.). All CPE isolates collected from 1 January 2022 to 31 December 2024 were reviewed. Only the first isolate per patient was considered. CFD susceptibility testing was performed during this period. All CPE isolates were recovered from human clinical or epidemiological samples submitted to the HUMS Laboratory as part of routine diagnostic procedures.

Inclusion criteria:(a)Strains belonging to the *Enterobacteriaceae* family;(b)Strains with carbapenemase detected by phenotypic or molecular methods;Strains with informed CFD susceptibility.

Exclusion criteria:(a)No archived strain available;(b)Non-viable archived strain;(c)Contaminated archived strain.

Bacterial strain archives were maintained in tryptone soy broth supplemented with 20% glycerol at −80 °C. Subcultures were performed on Columbia blood agar (Oxoid™ Thermo Fisher, Waltham, MA, USA) and incubated at 35 °C for 24 h to confirm species identity, validate CFD susceptibility, and increase biomass for whole-genome sequencing.

Species-level identification was performed following the manufacturer’s instructions using MALDI-TOF MS (Bruker Daltronics GmbH, Bremen, Germany). MALDI-TOF scores greater than 2.0 were considered valid.

CFD susceptibility testing was confirmed by disc diffusion on Müeller–Hinton agar using the archived isolates after confirming their identification. A second subculture was carried out on blood agar and incubated for 18–24 h at 35 ± 1 °C under aerobic conditions. A 0.5 McFarland suspension was prepared in 0.9% saline and evenly spread onto the surface of Müeller–Hinton agar using a sterile swab. A Cefiderocol 30 ug disc (Oxoid™ Cefiderocol disc, Oxoid Ltd., Wade Road, Basingstoke, Hampshire, RG24 8PW, United Kingdom) was placed at the centre of the plate, and was then incubated under aerobic conditions at 35 ± 1 °C. Results were read after 18 ± 2 h and interpreted according to the European Committee on Antimicrobial Susceptibility Testing breakpoint tables for the interpretation of MICs and zone diameters, version 15.0, 2025 [40].

Complementary antibiotic susceptibility testing was performed using the broth microdilution method with a MicroScan™ WalkAway semi-automated system (Beckman Coulter, Brea, CA, USA). Results were interpreted according to the European Committee on Antimicrobial Susceptibility Testing breakpoint tables for the interpretation of MICs and zone diameters, version 15.0, 2025.

CPE isolates from clinical samples were identified either by immunochromatographic assays (NG-Test^®^ CARBA 5, NG-Biotech Laboratories, Guipry-Messac, France) or by genotypic methods using isothermal amplification (Eazyplex^®^, Amplex Diagnostics GmbH, Gars am Inn, Germany) or a FilmArray^®^ system (BioFire Diagnostics LLC, Salt Lake City, UT, USA), depending on the case and in accordance with the laboratory’s internal protocols. All strains that tested positive by molecular methods for carbapenemase genes were classified as CPE, regardless of their minimum inhibitory concentration to carbapenems.

Epidemiological samples were initially screened for carbapenem resistance using Brilliance™ CRE chromogenic selective medium (Oxoid Limited, Basingstoke, UK). Confirmation of presumptive CPE was achieved via NG-Test^®^ CARBA 5 or real-time PCR using an Xpert^®^ Carba-R assay (Cepheid, Sunnyvale, CA, USA).

All identified CPE isolates with a pure and viable archived culture were eligible for WGS. Genomic DNA was extracted from 60 to 80 isolated colonies using a magnetic capture-based protocol with a MagCore^®^ system (RBC Bioscience, New Taipei City, Taiwan), following the manufacturer’s instructions, yielding 60 μL of eluate. Sequencing libraries were prepared using a Nextera XT™ DNA Library Prep Kit (Illumina Inc., San Diego, CA, USA). DNA concentration and quality were assessed throughout the process using Qubit™ fluorometric quantification (Thermo Fisher Scientific, Inc.) and Bioanalyzer™ analysis (Agilent Technologies, Inc., Santa Clara, CA, USA). Samples with DNA concentrations below 2 ng/μL or insert size distributions outside the 300 ± 50 bp range were excluded.

Sequencing was performed on an Illumina^®^ MiSeq™ platform using MiSeq V2 300-cycle reagent kits (Illumina Inc., San Diego, CA, USA), applying a 150 bp paired-end protocol with a targeted average sequencing depth greater than 50X. Libraries were loaded at a concentration of 12.5 pM, including 5% PhiX as a sequencing quality control.

De novo genome assembly was performed using Unicycler v0.5.1 [41]. The structural and functional quality of the assemblies was assessed using QUAST v5.2.0 [42] and BUSCO v5.6.1 [43], respectively. Assemblies were excluded based on the following quality criteria: fewer than 800,000 reads with a median Phred quality score below 28, more than 10% undetermined bases (Ns), an N50 value less than 30,000, or an assembled genome size falling outside the expected range of 5.5 ± 1.5 Mb. Contamination in raw reads and assembled genomes was evaluated using Mash v2.3 [44] and GUNC v1.0 [45], respectively. Final assembly graphs were manually reviewed using Bandage v0.8.1 [46]. Data were submitted to GenBank on 15 May 2025 under Submission ID SUB15324465. Sequences can be found under BioProject accessions PRJNA1263540 and PRJNA1190923.

Species-level identification was confirmed using GAMBIT v1.0.1 [47] and the PubMLST online species identification tool [48] (https://pubmlst.org/species-id, accessed on 1 February 2025). Multi-locus sequence typing was performed using mlst v2.23.0 [49], except for *K. pneumoniae*, which was analyzed separately. For *Klebsiella* species, MLST typing, resistance, and virulence gene annotation were conducted using Kleborate v3.1.3 [50]. Structural genome annotation was performed using Prokka v1.14.6 [51], and functional annotation was conducted with Sma3s v2 [52] using the UniRef90 database. Resistance genes were identified using RGI v6.0.3 against the CARD v4.0.0 database [53]. Resistance determinants with >80% coverage and >95% identity were retained for analysis.

Plasmidome assembly and characterization were performed using MOB-suite v3.1.9 [54]. Further annotation of plasmid content was conducted using RGI v6.0.3 (CARD v4.0.0), and visualization of plasmidome was carried out using Proksee (https://proksee.ca/, accessed on 5 March 2025).

The pangenome was calculated separately for resistant and susceptible isolates of *K. pneumoniae*, *E. coli*, and *E. hormaechei* using Roary v3.11.2 [55]. Presence/absence analysis was carried out with scoary v1.6.16 [56]. Results were visualized with Phandango v1.3.1 (https://jameshadfield.github.io/phandango/#/, accessed on 15 March 2025). Core genome alignment was performed with MAFFT v7.505 [57], and phylogenetic reconstruction was carried out using FastTree v2.1.11 [58].

Amino acid variant analysis of class C beta-lactamases and carbapenemases was performed in both groups by aligning the protein sequences (translated from nucleotide sequences) and manually inspecting them using MEGA12 v12.0.10 [59]. The reference sequences for each beta-lactamase are detailed in the Appendix A. Subsequently, only variants present in the resistant group and absent from the cefiderocol-susceptible group were selected. The root mean square deviation induced by the P209S substitution in the Ω-loop of EC-8 was calculated using PyMOL v3.1.4.1 (Schrödinger, LLC. The PyMOL Molecular Graphics System, 2015) based on the predicted three-dimensional structure of EC-8 obtained from SWISS-MODEL (https://swissmodel.expasy.org/interactive, accessed on 20 March 2025).

The truncation levels of OmpK35 and OmpK36 and the presence of GD and TD deletions in OmpK36 from the *K. pneumoniae* isolates were assessed by extracting and translating the respective gene sequences from the annotations, aligning them to their homologs in the reference strain *Klebsiella pneumoniae* subsp. *pneumoniae* HS11286, and performing manual inspection using the tools described above.

Single nucleotide variant (SNV) detection was performed via mapping and variant calling using Snippy v4.6.0 [60]. Detected variants were annotated with SnpEff v5.0 [61]. Custom in-house scripts were used to filter and retain all non-synonymous variants located in genes related to iron metabolism, PBPs, and the efflux systems *acr*, *tolC*, and *oqx*. Further filtering retained only the variants exclusively present in the resistant isolates and absent from the susceptible ones. The reference genomes used for each species are listed in the Appendix A.

ΔΔG energy shifts caused by the identified nucleotide variants were calculated using the BuildModel function from FoldX v5.0 [62], using as a reference the three-dimensional structures of the corresponding proteins from *Escherichia coli* K12, as predicted by the AlphaFold Protein Structure Database (https://alphafold.ebi.ac.uk/, accessed on 22 March 2025).

Inferential statistical analyses were conducted using various functions from the *rstatix* v0.7.2 package in R v4.4.2.

All nucleotide variants included in this study are either missense or nonsense mutations. All reported variants are presented as inferred amino acid substitutions to ensure consistency and comparability.

## 6. Conclusions

This study reinforces the concept that cefiderocol resistance is multifactorial in nature, identifying multiple loci that are potentially involved in a variety of clinical isolates from human-derived bacteria. The detection of exclusive mutations in siderophore-related genes with predicted structural impacts supports the hypothesis that the evasion of the drug’s “trojan horse” entry mechanism is a key driver of resistance, particularly through alterations in the *fec*, *fhu*, and *cir* operons, as well as the presence of specific beta-lactamases, notably NDM-type metallo-beta-lactamases.

Other elements, such as modifications in penicillin-binding proteins (PBPs), altered membrane permeability (e.g., loss of OmpK35/36), and the upregulation of efflux systems like AcrAB may act as complementary resistance mechanisms.

Despite the inherent challenges posed by multifactorial antibiotic resistance mechanisms, these findings hold substantial translational potential for clinical practise. They highlight several candidate targets and narrow the molecular focus for further investigation into their impact on cefiderocol efficacy, while also identifying possible substrates for the development of rapid molecular diagnostic tests. This is particularly relevant in the growing context of multidrug resistance, contributing to the development of tools that support targeted antimicrobial therapy and reinforcing antimicrobial stewardship programmes.

We are likely entering a new era in the study of antimicrobial resistance, in which traditional resistance models based on single genes are giving way to more complex, multifactorial mechanisms. This highlights the value of omics-based approaches to better understand emerging bacterial resistance pathways and to guide the development of new countermeasures. Future efforts should move beyond the resistome and incorporate comprehensive genomic surveillance and functional validation platforms in the study of multidrug-resistant pathogens, while also encouraging the engagement of clinical teams dedicated to the management of difficult-to-treat infections.

## Figures and Tables

**Figure 1 antibiotics-14-00703-f001:**
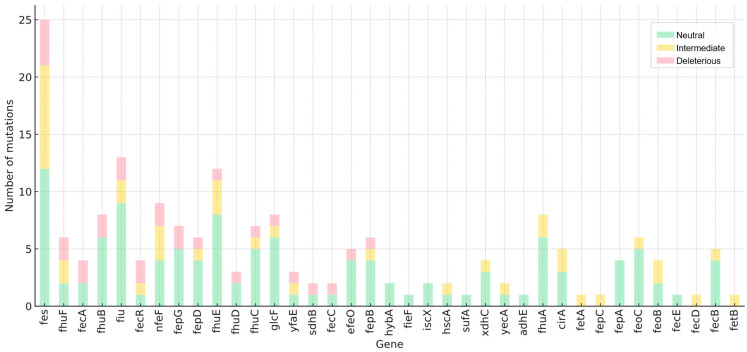
Number of missense mutations by impact type per gene.

**Figure 2 antibiotics-14-00703-f002:**
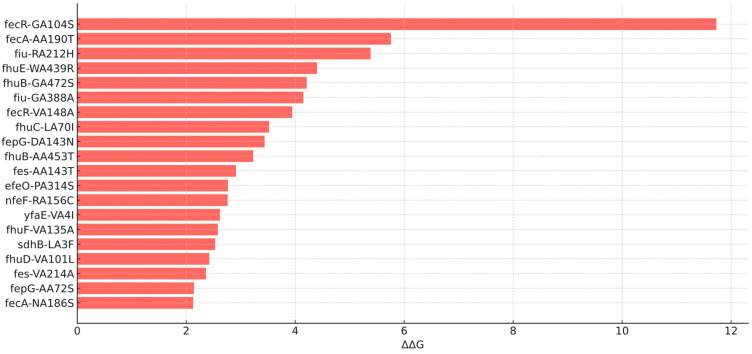
Top 20 deleterious aminoacid changes by calculated delta energy gap.

**Figure 3 antibiotics-14-00703-f003:**
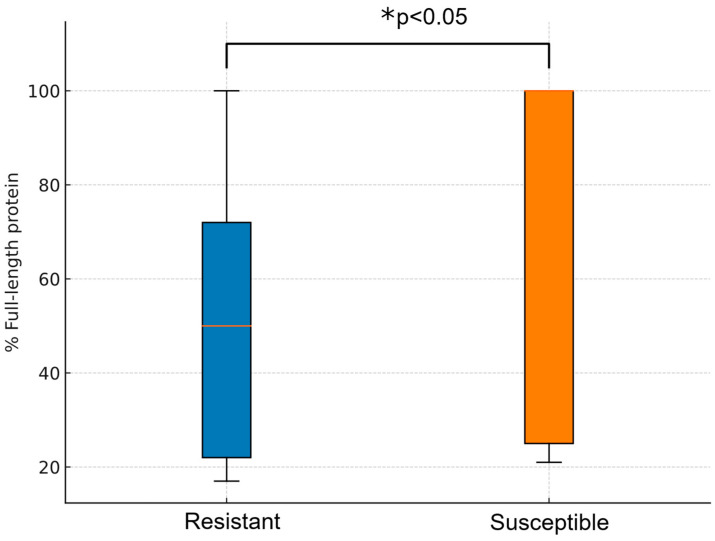
Comparison of full-length OmpK35 between groups. In the boxplot, the whiskers represent the range of values, while the boxes correspond to the interquartile range (25th–75th percentile). The resistant group exhibits a significantly shorter predicted OmpK35 protein length, suggesting a potential loss of porin function.

**Figure 4 antibiotics-14-00703-f004:**
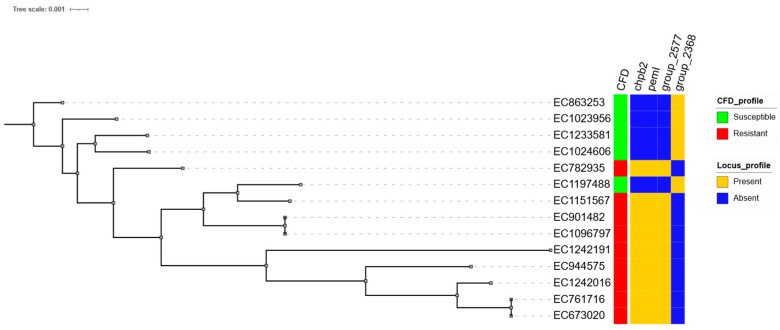
Core genome base phylogeny of *E. coli* strains and its CFD and exclusive locus profile. Most phylogenetically related isolates appear to share a specific resistance profile, along with the presence of *chpB2*, *pemI*, and group_2577, and the absence of group_2368. This pattern may be associated with potential surrogate markers of the phenotypic resistance profile.

**Table 1 antibiotics-14-00703-t001:** Microbiological and genomic profiles of the isolates.

Sample Name	CFD Profile	Sample Type	Species Name	ST	Carbapenemase	ESBL	AmpC	Other Beta-Lactamases	Plasmid Size
CF1313021	R	Triple-swab	*Citrobacter portucalensis*	1129	NDM-1	0	CMY-13	0	-
EC1096797	R	Triple-swab	*Escherichia coli*	167	NDM-1; OXA-244	CTX-M-15	EC-15	TEM-1	219.5
EC1151567	R	Triple-swab	*Escherichia coli*	361	NDM-5; KPC-3	CTX-M-15	CMY-145	OXA-1;SHV-11;TEM-1	78.1
EC1242016	R	Rectal swab	*Escherichia coli*	38	NDM-1	CTX-M-15	CMY-16; EC-8	OXA-1; OXA-10	246.8
EC1242191	R	Triple-swab	*Escherichia coli*	131	OXA-48	CTX-M-163	CMY-181	OXA-1;TEM-1	4.3
EC673020	R	Surgical wound	*Escherichia coli*	2659	NDM-5	0	CMY-42; EC-8	TEM-1	120.1
EC761716	R	Urine	*Escherichia coli*	2659	NDM-5	0	CMY-42; EC-8	TEM-1	119.3
EC782935	R	Triple-swab	*Escherichia coli*	410	OXA-181	0	CMY-4; EC-14	OXA-1	27.9
EC901482	R	Urine	*Escherichia coli*	167	NDM-1; OXA-244	CTX-M-15	EC-15	TEM-1	201.9
EC944575	R	Triple-swab	*Escherichia coli*	405	NDM-5	CTX-M-55	EC-8	OXA-1;TEM-1	79.4
EE1280220	R	Triple-swab	*Enterobacter hormaechei*	108	OXA-181	0	ACT-55	0	49.8
EE1332654	R	Urine	*Enterobacter kobei*	191	VIM-1	0	ACT-52	0	96.9
EE1338763	R	Ascitic fluid	*Enterobacter asburiae*	702	OXA-48	0	MIR-3	0	-
EE974926	R	Wound	*Enterobacter hormaechei*	51	KPC-3	0	ACT-40	TEM-1	50.6
KP1059745	R	Triple-swab	*Klebsiella pneumoniae*	307	KPC-2; NDM-1	0	0	SHV-28	95.1
KP1067518	R	Rectal swab	*Klebsiella pneumoniae*	23	NDM-1;OXA-48	CTX-M-55	0	SHV-1	66.9
KP1207364	R	Triple-swab	*Klebsiella pneumoniae*	101	KPC-3	0	0	SHV-1	15.4
KP1207896	R	Wound	*Klebsiella pneumoniae*	395	NDM-1; OXA-48	0	CMY-6	SHV-11	353.8
KP1207904	R	Triple-swab	*Klebsiella pneumoniae*	512	KPC-3	0	0	SHV-11	-
KP1215397	R	Surgical wound	*Klebsiella pneumoniae*	392	NDM-1	CTX-M-15	0	SHV-11;TEM-1	342.5
KP1234533	R	Urine	*Klebsiella pneumoniae*	395	NDM-1; OXA-48	CTX-M-15	0	OXA-1;SHV-1;TEM-257	339.2
KP1268564	R	Abscess	*Klebsiella pneumoniae*	14	VIM-1	0	0	SHV-1	76.7
KP1307832	R	Urine	*Klebsiella pneumoniae*	11	KPC-3	0	0	OXA-1;SHV-11	-
KP944560	R	Triple-swab	*Klebsiella pneumoniae*	23	NDM-1; OXA-48	CTX-M-55	0	SHV-1	6.6
KP944575	R	Triple-swab	*Klebsiella pneumoniae*	405	NDM-1	CTX-M-15	0	OXA-1;SHV-28; TEM-1	342.9
KP953369	R	Triple-swab	*Klebsiella pneumoniae*	147	NDM-1; OXA-48	CTX-M-15	0	OXA-1; OXA-9;SHV-1	99.3
KP971943	R	Triple-swab	*Klebsiella pneumoniae*	395	NDM-1	CTX-M-15	0	OXA-1;SHV-11 35Q; TEM-1	338.9
KP985068	R	Surgical wound	*Klebsiella pneumoniae*	512	KPC-3	0	0	SHV-11 35Q	48.5
PS965060	R	Urine	*Providencia stuartii*	405	NDM-5	0	0	0	8.1
CF775268	S	Rectal swab	*Citrobacter portucalensis*	493	VIM-1	CTX-M-9	CMY-2	OXA-1	38.0
CK1116243	S	Rectal swab	*Citrobacter koseri*	937	VIM-24	CTX-M-9	CKO-1	OXA-1	290.0
EC1023956	S	Rectal swab	*Escherichia coli*	29	VIM-1	0	EC-14	TEM-1	7.7
EC1024606	S	Rectal swab	*Escherichia coli*	539	VIM-1	0	EC-18	0	115.3
EC1197488	S	Triple-swab	*Escherichia coli*	409	KPC-3	0	EC-15	SHV-11	52.4
EC1233581	S	Urine	*Escherichia coli*	602	NDM-5	CTX-M-15	EC-15	TEM-1	41.1
EC863253	S	Rectal swab	*Escherichia coli*	327	VIM-1	0	EC-14	0	24.5
EE1274028	S	Rectal swab	*Enterobacter hormaechei*	45	VIM-1	0	0	SHV-12;TEM-1	87.2
EE1318769	S	Rectal swab	*Enterobacter hormaechei*	90	VIM-24	CTX-M-9	ACT-56	OXA-1	320.3
KP1045007	S	Rectal swab	*Klebsiella variicola*	4365	VIM-24	CTX-M-9	0	OXA-1;LEN-16	289.8
KP1096796	S	Triple-swab	*Klebsiella pneumoniae*	147	NDM-1;OXA-48	CTX-M-15	0	OXA-1; OXA-9;SHV-1; TEM-1	112.8
KP1096799	S	Triple-swab	*Klebsiella pneumoniae*	147	NDM-1;OXA-48	CTX-M-15; TEM-150	0	OXA-1; OXA-9;SHV-1	49.2
KP1131939	S	Surgical wound	*Klebsiella pneumoniae*	3817	VIM-1	0	DHA-1	SHV-1	99.5
KP1156073	S	Urine	*Klebsiella pneumoniae*	15	OXA-48	CTX-M-15	0	OXA-1;SHV-28;TEM-1	61.4
KP1174934	S	Prosthetics	*Klebsiella pneumoniae*	395	OXA-48	CTX-M-15;CTX-M14	0	SHV-11;TEM-1	74.9
KP1216215	S	Urine	*Klebsiella pneumoniae*	9	VIM-1	0	0	SHV-161	12.2
KP1255048	S	Ear swab	*Klebsiella pneumoniae*	395	NDM-1;OXA-48	CTX-M-15	0	OXA-1;SHV-11;TEM-1	360.5
KP1289033	S	Urine	*Klebsiella pneumoniae*	307	NDM-1	0	0	SHV-28	-
KP1348849	S	Triple-swab	*Klebsiella pneumoniae*	20	VIM-1	0	0	SHV-187	228.3
KP822390	S	Triple-swab	*Klebsiella pneumoniae*	147	NDM-1	CTX-M-15; TEM-150	0	OXA-1; OXA-9;SHV-1	95.0
KP838840	S	Blood culture	*Klebsiella pneumoniae*	4387	VIM-1	0	0	SHV-1	71.7
KP844839	S	Surgical wound	*Klebsiella pneumoniae*	395	NDM-1;OXA-48	CTX-M-15; TEM-105	0	OXA-1;SHV-1	338.5
KP846745	S	Rectal swab	*Klebsiella pneumoniae*	147	NDM-1;OXA-48	CTX-M-15; TEM-150	0	OXA-1; OXA-9;SHV-1;	99.4
KP882410	S	Triple-swab	*Klebsiella pneumoniae*	307	NDM-1	CTX-M-15	0	OXA-1;SHV-28; TEM-1	337.3
KP896137	S	Rectal swab	*Klebsiella pneumoniae*	584	VIM-1	0	0	OXA-1;SHV-168	61.9
KP932969	S	Rectal swab	*Klebsiella pneumoniae*	268	VIM-1	0	DHA-1	SHV-1; DHA-1	80.3
MM1207184	S	Skin ulcer	*Morganella morganii*	-	NDM-1	CTX-M-15	0	TEM-1	179.9
PR1307361	S	Urine	*Providencia hangzhouensis*	44	NDM-1	0	0	0	-
PS1207364	S	Triple-swab	*Providencia stuartii*	11	NDM-5	0	0	0	-
SM1131939	S	Wound	*Serratia sarumanii*	522	VIM-1	0	SRT-2	0	12.3

S = Susceptible; R = resistant. Plasmid sizes are expressed in kilobases.

**Table 2 antibiotics-14-00703-t002:** Variants per locus exclusive of the CFD-R group.

Resistance Mechanism	Locus	Amino Acid Variant
Iron metabolism	*cirA*	D95G, E465D, E507fs, I174V, I547F, I547L, R514fs
	*fecB*	A134T, A214S, D55Y, I57S, L8V, T23M
	*fes*	A143T, A189V, A264D, A272G, A327T, D99V, E192G, E329Q, EY42GH (complex), H293N, I163T, I163T (complex), I343V, I362L, I53V, K177N, K324Q, K324Q(complex), L130P, L261Q, N75D (complex), P164A, Q222R, Q316H, Q66R, R174W, R350Q, T186I, T45A, T45P, T80K, V104A, V214A (complex), V30I, V320M, V51I, V56M
	*fhuC*	A122V, A72T, E239D, E67A, L70I, M100L, S188A, S188A (complex), S64T
	*fhuF*	A127V, A208T, A64T, C214Y, D176G, E119Q, E144D, H155Q, I179N, K149R, K35E, L214I, L55Q, M83T, P52L, P63T, PT23AG (complex), QDPT21HDAG (complex), R126C, S153A, S163R, SQ58TE (complex), T219M, V12I, V135A, V135A (complex), V65A
	*fiu*	A417T, D70N, G388A, M513V, Q58K, R212H, S389A, T367A, T38A, T493A, T493A (complex), V211A, V235I, V495M, V630M, Y274F
	*nfeF*	A172V, A237T, D107E, DG107ED (complex), G113C, G113S, N179H, P61S, P81S, Q119K, R156C, R24H, T4S, V25A, Y240F
PBP	*dacB*	A121T, K112R, L136F, P182Q, R228S, T269A, V18I
	*ftsI*	A233T, E349K, I332V, I532L, Q227H
	*mrcA*	A373V, G414D, R711H, S497G
	*mrcB*	D765N, H604N, R556C
	*mrdA*	A530S, D354N
Efflux pumps	*TolC*	E230D, G243D, I354L, I3M, L8I, M5I, N212D, N28S, N31S, N436S, Q167K, Q169K, Q356R, Q429L, R289S, S124G, S313A, T61R, V165I, V328I, V49A, LA30QT (complex)
	*acrA*	E142D, L147Q, M334T, S122A, S73N, T104A, T379K
	*acrB*	H596N, K1035N, S1043N
	*acrD*	A28T, A696T, D308E, I841V, K652E, L230V, N248D, N74D, N793S, S804T, T851A, V1026I, V575I
	*acrE*	D327N, N103S, N77S, P302S, Q260P, R167H, T382F, T382S
	*acrF*	A24V, E429D, H338Q, K428R, K849Q, S806A
	*acrR*	A117T, A145S, A146T, A163T, A183T, A20D, A45V, A7T, A80T, D11E, D157V, E186T, E196D, E79D, E91A, F38L, G115S, G168C, G78S, K193Q, K56R, L58V, N130S, P206L, P216S, Q139H, Q141K, Q191K, Q64H, R135H, R13C, R176K, R23K, R62C, R9H, S116N, S120Y, S184T, S85P, T183A, T54N, T73A, V101A, Ala47fs, K80fs, L101fs, L109fs, V29fs, QS152RT (complex), LS212HN, TN213IT, Q122*
	*oqxB*	A203T, A851V, D1046E, N798S, FA550IV
	*tolC*	A233T, E205Q, I280V, K139N, L108M, N137Q, N489T, S467G, S476P, T483A
Beta-lactamases	*bla_EC-14_*	Q23K, R248C, H312R, A367T
	*bla_EC-15_*	Q23K, P110S, A367T
	*bla_EC-8_*	T4M, D140E, N201T, P209S, S298I, T321A, T367A
	*bla_ACT-55_*	V311dup
Porin loss	*OmpK36*	TD134ins

## Data Availability

The data were submitted to GenBank on 15 May 2025 under Submission ID SUB15324465. The sequences can be found under BioProject accession PRJNA1263540 and PRJNA1190923.

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
