# Peer review of "Mechanisms of Cefiderocol Resistance in Carbapenemase-Producing Enterobacterales: Insights from Comparative Genomics"

_antibiotics, 2025, doi:10.3390/antibiotics14070703_

Round 1
Reviewer 1 Report
Comments and Suggestions for Authors
Major comments:
- the discussion lacks critical engagement with the findings; authors repeat observed results without discussing why changes might have occurred
-the conclusions need some description of clinical impact of the data
Minor comments:
- use „Enterobacterales” instead „Enterobacteriacae” – this systematic change is well-established in literature (e.g. line 21); check whole manuscript
- do not italicize „spp.” (e.g. line 74); check whole manuscript
- cite EUCAST Tables ver. 15 properly (lines 488-489)
Reviewer 2 Report
Comments and Suggestions for Authors
This is indeed a very well written and well researched manuscript where the authors (investigators) have done an extensive study to identify the genomic factors responsible for developing resistance towards the Cefiderocol, a very effective cephalosporin antibiotic used against many multi-drug resistant Gram-negative bacterial pathogens. The study led to generate a useful database in the context of Cefiderocol resistance in hospital acquired infections. Though the authors could not single out any specific factor responsible for developing resistance in multiple species of bacterial isolates, they have identified many probable factors which might be responsible for resistance development. Authors have emphasized on the fact that it is a multi-factorial phenomenon. Overall, the study has been designed very well and it is a good amount of work to be published. However, there are some weak links of the work which authors have also acknowledged in the manuscripts. Here are the list of things which could be be improved in the manuscript.
- The sample size for the study should be much higher than the number of isolates used in this study. Among those small number of isolates, there were multiple species which made it more statistically challenging to make any interpretation.
- Inter-species genomic variability can be an important factor in the study. Authors may put some light on it.
- Figure 3 & 4 legends could be more descriptive for the understanding of the readers.
- It would be interesting to see the antimicrobial profiles of these isolates particularly against some other cephalosporins and beta-lactam antibiotics.
- How does the mutation in the efflux pumps confer resistance towards the cephalosporins?
Reviewer 3 Report
Comments and Suggestions for Authors
The manuscript addresses a topic of great interest to public health and presents a well-structured and scientifically sound study. The introduction is clear and concise, effectively setting the stage for the research objectives. The methodology is well described and appears to be reproducible, with appropriate detail provided. The results are consistent with the methods applied and are presented in a logical and coherent manner. Both the discussion and the conclusions are appropriately based on the data generated, reinforcing the strength and relevance of the study.
However, there are several points that require attention. In Table 1, the column headings contain underscores ( _ ), which should be removed to maintain proper formatting and readability. Furthermore, there appears to be an inconsistency in the classification of ESBLs. The authors list SHV-11 under other β-lactamases, despite the fact that this, as other variants mentioned (e.g., SHV-28, TEM-105, TEM-150) are recognised ESBLs. It seems that only CTX-M producers are being identified as true ESBLs, which is not accurate. The authors are encouraged to revise this classification to more accurately reflect the diversity of ESBL enzymes.
In the abstract (line 21), there is a closing parenthesis that does not have a corresponding opening parenthesis; this should be corrected. Additionally, in line 192, E. coli must be italicised, as should K. pneumoniae in line 414. All bacterial species names throughout the manuscript must be presented in italics, in accordance with standard scientific conventions.
Overall, the manuscript makes a valuable contribution, but a number of formatting and classification issues should be addressed to ensure scientific accuracy and editorial quality.
